# Efficacy, Pharmacokinetics, and Toxicity Profiles of a Broad Anti-SARS-CoV-2 Neutralizing Antibody

**DOI:** 10.3390/v15081733

**Published:** 2023-08-14

**Authors:** Silvia Godínez-Palma, Edith González-González, Frida Ramírez-Villedas, Circe Garzón-Guzmán, Luis Vallejo-Castillo, Gregorio Carballo-Uicab, Gabriel Marcelín-Jiménez, Dany Batista, Sonia M. Pérez-Tapia, Juan C. Almagro

**Affiliations:** 1Unidad de Desarrollo e Investigación en Bioterapéuticos (UDIBI), Escuela Nacional de Ciencias Biológicas, Instituto Politécnico Nacional, Mexico City 11340, Mexico; silvia.godinez@udibi.com.mx (S.G.-P.); edith.gonzalez@udibi.com.mx (E.G.-G.); frida.ramirez@udibi.com.mx (F.R.-V.); circe.garzon@udibi.com.mx (C.G.-G.); luis.vallejo@udibi.com.mx (L.V.-C.); gregorio.carballo@udibi.com.mx (G.C.-U.); 2Laboratorio Nacional Para Servicios Especializados de Investigación, Desarrollo e Innovación (I+D+i) Para Farmoquímicos y Biotecnológicos, LANSEIDI-FarBiotec-CONACyT, Mexico City 11340, Mexico; 3Pharmometrica Analytical & Statistics Unit, Av. Eje 5 Norte 990, Edificio “C” planta baja, Mexico City 02230, Mexico; gabmarcelin@pharmometrica.com.mx (G.M.-J.); dbatista@pharmometrica.com.mx (D.B.); 4Departamento de Inmunología, Escuela Nacional de Ciencias Biológicas, Instituto Politécnico Nacional (ENCB-IPN), Mexico City 11340, Mexico; 5GlobalBio, Inc., 320 Concord Ave, Cambridge, MA 02138, USA

**Keywords:** COVID-19, therapeutic antibody, SARS-CoV-2 Delta, SARS-CoV-2 Omicron, toxicology

## Abstract

We recently reported the isolation and characterization of an anti-SARS-CoV-2 antibody, called IgG-A7, that protects transgenic mice expressing the human angiotensin-converting enzyme 2 (hACE-2) from an infection with SARS-CoV-2 Wuhan. We show here that IgG-A7 protected 100% of the transgenic mice infected with Delta (B.1.617.2) and Omicron (B.1.1.529) at doses of 0.5 and 5 mg/kg, respectively. In addition, we studied the pharmacokinetic (PK) profile and toxicology (Tox) of IgG-A7 in CD-1 mice at single doses of 100 and 200 mg/kg. The PK parameters at these high doses were proportional to the doses, with serum half-life of ~10.5 days. IgG-A7 was well tolerated with no signs of toxicity in urine and blood samples, nor in histopathology analyses. Tissue cross-reactivity (TCR) with a panel of mouse and human tissues showed no evidence of IgG-A7 interaction with the tissues of these species, supporting the PK/Tox results and suggesting that, while IgG-A7 has a broad efficacy profile, it is not toxic in humans. Thus, the information generated in the CD-1 mice as a PK/Tox model complemented with the mouse and human TCR, could be of relevance as an alternative to Non-Human Primates (NHPs) in rapidly emerging viral diseases and/or quickly evolving viruses such as SARS-CoV-2.

## 1. Introduction

Coronavirus disease 2019 (COVID-19), caused by the severe acute respiratory syndrome coronavirus 2 (SARS-CoV-2), prompted an unprecedented search for diagnostic, prophylactic, and therapeutic solutions for controlling this devastating worldwide pandemic. In response to this call and due to the success of antibodies in preventing and treating diverse infectious diseases [1,2], hundreds of small, medium, and large biotech companies, as well as academic laboratories, focused their efforts on isolating and characterizing anti-SARS-CoV-2 antibodies. As a result, nine therapeutic and/or prophylactic antibodies specific for the receptor-binding domain (RBD) of SARS-CoV-2 received Emergency Use Authorization (EUA) from the Food and Drug Administration (FDA) and/or European Medicines Agency (EMA) during the first two years of the pandemic [1].

The EUA antibodies initially demonstrated safety and efficacy in human clinical studies for the original variant of the virus, known as the Wuhan (WT) strain. However, as new immune scape SARS-CoV-2 mutants or Variants of Concern (VOCs) emerged, almost all the EUA antibody-based drugs partially lost their efficacy or became non-efficacious against the VOCs, in particular Omicron (B.1.1.529 lineage) and its sub-variants [1,2,3,4]. This led to a race for the quick development of efficacious therapeutic antibodies with a broad neutralization profile to mitigate the further spread of SARS-CoV-2 by treating or preventing infections with new and/or reemergent VOCs, including the implementation of creative solutions for discovery, efficacy studies, toxicology testing, and COVID-19 clinical trials.

In a previous work [5,6], we reported the isolation of an antibody called IgG-A7 that neutralized the WT, Delta (B.1.617.2), and Omicron strains in plaque reduction neutralization tests (PRNTs) with authentic viruses. IgG-A7 also protected transgenic K18-hACE2 mice for the hACE-2 from a SARS-CoV-2 WT infection at a dose of 5 mg/kg [6]. In this work, we expanded the efficacy studies of IgG-A7 in K18-hACE2 mice infected with Delta and Omicron at doses of 0.5 and 5 mg/kg. Further, we studied the pharmacokinetic (PK) profile and toxicity (Tox) of IgG-A7 in CD-1 mice at single doses of 100 and 200 mg/kg, which are 20- and 40-fold higher than the highest efficacious dose of 5 mg/kg. The PK profile of IgG-A7 was consistent with the PK parameters of human IgG1 antibodies when using mice as a species for PK/Tox studies. Further, IgG-A7 was well tolerated and showed no signs of toxicity. Finally, a Tissue Cross-Reactivity (TCR) study of IgG-A7, with a panel of 13 and 34 relevant mouse and human tissues, respectively, showed no interaction of IgG-A7 with the tissues of these species, supporting the PK/Tox results and suggesting that this broadly neutralizing anti-SARS-CoV-2 antibody could be well-tolerated in humans. The lessons learned by using CD-1 mice as a Tox species for rapidly evolving viruses such as SARS-CoV-2 are discussed.

## 2. Materials and Methods

### 2.1. Animals and Ethics

The efficacy studies were performed on groups of male and female K18-hACE2 transgenic mice for hACE2; B6.CgTg (K18ACE2)2Prlmn/JHEMI strain number 034860, aged 6 to 8 weeks, purchased from Jackson Laboratories, USA. The PK and Tox studies were performed with male and female CD-1 (Crl: CD1 ICR) mice aged 6 to 10 weeks, purchased from the Unidad de Producción de Animales de Laboratorio (UPEAL) of the Universidad Autónoma Metropolitana (UAM). All the mice were maintained in quarantine for 1 week before each study and showed no signs of disease or illness upon arrival and before the administration of the treatments. During quarantine, efficacy and PK/Tox studies the temperature of the animal facility was maintained at 20–25 °C and the humidity at 50–75%, with a minimum of eight air changes per hour. The animals were kept on a 12 h light/dark cycle and had free access to water and food (LabDiet, St. Louis, MO, USA, Cat. 5010).

All the studies were approved by the following committees: (1) The Biosafety Comité del Instituto de Oftalmología Fundación de Asistencia Privada Conde de la Valenciana IAP; (2) Comité Interno para el Cuidado y Uso de Animales de Laboratorio (CICUAL) de la CPA del Servicio Nacional de Sanidad, Inocuidad y Calidad Agroalimentaria (SENASICA); and (3) Comité de Ética e Investigación de la ENCB (CEI-ENCB). The animals were treated and the experiments were performed according to the regulations outlined in the NOM-062-ZOO-1999 [7].

### 2.2. Efficacy of IgG-A7 against SARS-CoV-2 Delta and Omicron

The SARS-CoV-2 Delta and Omicron viruses were isolated and characterized as described in González-González [6]. The whole genome sequences of these SARS-CoV-2 strains are deposited in GenBank with the following accession numbers: OM060237 (Delta) and ON651664 (Omicron). SARS-CoV-2 was propagated in Vero E6 cells (ATCC, Cat. CRL-1586) for 60 h and frozen for one day to lyse the cells. The supernatants were collected and titrated with a plaque assay using Vero E6 cells [8] to generate working stocks. The SARS-CoV-2 stocks were diluted in EMEM to obtain 10^3^ PFU/40 µL and 10^5^ PFU/40 µL for the Delta and Omicron variants, respectively.

The virus was intranasally administered, with the mice being previously anesthetized with 65 mg/kg of Ketamine and 13 mg/kg of Xylazine. IgG-A7 was given intraperitoneally (i.p.) in a single dose of 0.5 or 5 mg/kg 24 h post-infection to groups of ten mice per dose (five male and five female). Their survival, body weight, and clinical signs were monitored daily for 14 days for infection with Delta. Their clinical signs and body weight were monitored for 7 days for infection with Omicron. The viral load in the lung was determined via RT-PCR to detect SARS-CoV-2 E gene copies on day 14 post-infection. The mice were euthanized when they showed irreversible signs of disease. Active virus manipulations, as well as efficacy studies, were performed at BSL2+ facilities, with strict biosafety standards and risk assessment protocols according to the specifications of the WHO Laboratory Biosafety Manual, Four Edition and the CDC Guidance for General Laboratory Safety Practices during the COVID-19 Pandemic [9,10,11,12].

### 2.3. PK Study at High Single Doses

Seventy-two CD-1 mice, gender balanced, were divided in two groups and administered with IgGA7 intravenously (i.v.) at a single bolus dose of 100 or 200 mg/kg of IgG-A7. The mice in the group dosed with 100 mg/kg (body weight 22.54 g ± 2.48 g) received 2.25 ± 0.25 mg in 225 µL. For the group with the 200 mg/kg dose (body weight 25.24 g ± 2.30 g), the mice received 5.07 ± 0.46 mg in 243 µL. At 1, 2, 4, 8, 24, 48, 96, 144, and 336 h after administration, four mice (two female and two male) at each of the nine time points were sacrificed and blood samples were collected via cardiac puncture into BD Microtainer^®^ Blood Collection Tubes (Cat. 365967, BD, USA). All the blood samples were centrifuged at 10,000× *g* for 10 min at 4 °C to obtain serum. The serum samples were stored at −80 °C until analysis.

The concentration of IgG-A7 in the serum samples was measured using an ELISA adapted from a previously reported assay developed by our group for the detection of anti-SARS-CoV-2 antibodies in human serum samples [13]. Briefly, Nunc Maxisorp 96-well plates (Thermo Fisher Scientific; Waltham, MA, USA) were coated with 1 µg/mL of SARS-CoV-2 receptor-binding domain (RBD) overnight at 4 °C. The plates were washed with PBS-0.1% Tween (PBST), blocked with 3% skim milk in PBS-Tween 0.1% (MPBST) for 1 h at room temperature (RT), and washed with PBST. The serum samples, standard curves of IgG-A7, and negative controls (D1.3 anti-lysozyme antibody) were prepared in 1% milk on PBS (MPBS), and 100 µL was added to the plates and incubated for 1 h at room temperature (RT). The plates were washed with PBST and IgG-A7 bound was detected with a goat α-hIgG antibody HRP-conjugated (Abcam Cat. ab97225; Cambridge, UK). The assay was revealed with TMB substrate reagent (BD OptEIA, BD Biosciences, San Diego, CA, USA, Cat. 555214). The reaction was stopped with phosphoric acid 1 M (Abcam, Cambridge, MA, USA, Cat. AB19312) and the plates were read at 450 nm with a correction at 570 nm using a SpectraMax M3 microplate reader (Molecular Devices, LLC; San José, CA, USA). The method was validated, following the criteria established in the M10 ICH Harmonized Guideline [14], to ensure the reliability of the assay (See Appendix A).

The PK parameters were determined using Phoenix WinNonlin^®^ (version 7.0, Certara USA, Inc., Princeton, NJ, USA), programming an i.v. bolus administration and a single-compartment model. The following PK parameters were calculated: area under the serum concentration–time curve to the last sampling time (AUC), maximum observed serum concentration (C_max_), time of the maximum observed serum concentration (T_max_), serum elimination half-life (t_½_), clearance (Cl), and volume of distribution (Vd).

### 2.4. Tox Studies

Three groups of 10 CD-1 mice (5 males and 5 females) each were used for the Tox studies. The first group received a 100 mg/kg single i.v. dose of IgG-A7. The second group received a single i.v. dose of 200 mg/kg of IgG-A7. The third group (control group) did not receive the antibody. The body weight of the animals was monitored daily for 14 days post IgG-A7 administration. As a reference, prior to the administration of the antibody, the body weights of the mice in the three groups were measured. Fourteen days post-administration, urine (micturition technique) and blood (cardiac puncture) samples were collected for urinalysis, blood chemistry, and hematology. Immediately after euthanasia (by carbon dioxide aspiration), mice lungs, livers, spleens, hearts, and kidneys were collected and fixed in 10% formaldehyde for histopathological analyses.

All the analyses were performed by the MAULAB Veterinary Clinical Pathology Laboratory (Benito Juárez, Mexico City, Mexico, https://maulab.com.mx/; accessed on 31 December 2022). For the urinalysis, urine pools from three male or three female mice were analyzed. The physical appearance (color, and urine density) was assessed. Further, a chemical examination (pH, proteins, glucose, ketones, bilirubin, urobilinogen, Blood/Hb, erythrocytes, leukocytes, squamous epithelial cells, transitional epithelial cells, and cylinders) was determined. In addition, microscopic examination (crystals, bacteria, lipids, and others) was performed. For hematology, impedance (EXIGO H400) and wright-stained blood smears were performed to determinate: the hematocrit (HCT), hemoglobin (HGB), erythrocytes (RBC), Mean Corpuscular Volume (MVC), Mean Corpuscular Hemoglobin Concentration (MCHC), platelets (PLK), White Blood Cells (WBC), neutrophils, lymphocytes, and monocytes. The toxicological analysis included blood chemistry of the following parameters: glucose, urea, creatinine, cholesterol, triglycerides, total bilirubin, Alanine Transaminase (ALT), Aspartate Aminotransferase (AST), Alkaline Phosphatase (AP), Creatinine Kinase (CK), total proteins, albumin, globulins, A/G ratio, calcium, and phosphorus. Finally, the organs (lungs, hearts, kidneys, spleens, and livers) were processed for microscopic examination with sections of the kerosene block in the microtome (5 mm) and were stained with hematoxylin and eosin. The slide analysis was carried out at magnifications of 10× and 40×.

### 2.5. TCR in Mouse and Humans

IgG-A7 was conjugated with Fluorescein isothiocyanate (FITC) using the Fluoro Tag™ conjugation kit (Sigma-Aldrich Cat. FITC1-1KT; St. Louis, MO, USA), as described by the manufacturer. Conjugation was performed using a 1:1 Antibody: FITC molar ratio and the non-conjugated fluorophore was removed using a 50 kDa centrifugal filter (Merck Mil-lipore; Burlington, MA, USA) and PBS 1X (Sigma-Aldrich). The quality control of the conjugated IgG-A7-FITC consisted of determining the protein content using densitometry (>1.8 mg/mL), the monomeric content using Size-Exclusion Chromatography (>99%), identity using Mass Spectrometry (>1 FITC molecule per antibody structure), and relative potency determined using the formula [EC50 IgG-A7-FITC/EC50 IgG-A7] ∗ 100% in an ELISA plate coated with RBD protein (94–96%).

The TCR studies were performed by HistologiX Ltd. (BioCity, Nottingham, UK). The mouse TCR study was performed on 13 frozen tissues listed in Appendix A, obtained from the Tissue Micro-Array (TMA) of FDA standard normal tissues (Amsbio, Abingdon, UK, Cat. T6334701-2). On the other hand, the GLP human TCR assay was performed using 34 normal human tissues collected by surgical excision or during post-mortem examinations and supplied by Tissue Solutions Ltd. (Glasgow, UK) (Appendix A).

The Tissue Micro Array sections were air-dried and then fixed with Neutral Buffered Phormalin (NBF) for 15 s, whereas the human tissue sections were cut to 5–8 µm, picked up on SuperFrost Plus™ slides (Thermo Scientific; Waltham, MA, USA), air-dried for 60 min at room temperature, and stored at −80 °C. The tissue sections were incubated with IgG-A7-FITC at an optimal concentration (0.625 µg/mL) for the mice study, and at optimal (0.625 µg/mL), supra-optimal (1.25 µg/mL), and sub-optimal (0.078 µg/mL) concentrations in the human TCR assay. The HEK293T SARS-CoV-2 spike protein stable cell line (GenScript Cat. M00804; Piscataway, NJ, USA) and HEK293T cell line (Takara Cat. 632180; CA, USA) were used as positive and negative controls, respectively.

The visualization and semi-quantitative analysis of the immunohistochemistry (IHC) results were performed by PathCelerate Ltd. (Mill Lane; Goostrey, UK) using light microscopy. The binding of IgG1-A7-FITC to the tissues was detected using a rabbit anti-FITC HRP conjugated (Ancell Cat. 295-040; Stillwater, MN, USA). The isotype IgG1 control was run at a concentration of 0.625 µg/mL and a negative control (primary antibody omitted) was also included. A vimentin ICH assay was used for integrity testing in both assays. Blocking was performed with normal horse serum and the vimentin antibody (Abcam Cat. Ab92547; Cambridge, MA, USA) was diluted to 1:400 (concentration 0.68 μg/mL), followed by the quenching of endogenous peroxidases with 0.3% hydrogen peroxide in methanol. The intensity of the staining was reported as negative (0), minimal (1), mild (2), moderate (3), and marked (4), whereas the staining distribution was reported as no cell staining (-), very rare cell staining (+), rare cell staining (++), and occasional staining (+++).

## 3. Results

### 3.1. IgG-A7 Efficacy against Delta and Omicron SARS-CoV-2 Variants

K18-ACE2 mice were infected with SARS-CoV-2 Delta and Omicron and treated with 0.5 and 5 mg/kg of IgG-A7. All the mice infected with Delta and treated with IgG-A7 survived (Figure 1A), in contrast to the infected but untreated mice, where all the mice died between day 6 and 9 post-infection. The body weights (Figure 1B) of the untreated animals decreased by 23% compared to the group without infection, while the groups treated with IgG-A7 did not present any weight loss, instead gaining around 6% with respect to their weight at the beginning of the study, similar to the uninfected control group. The viral load in the lungs (Figure 1C) of the mice infected with Delta and treated with doses of 0.5 or 5 mg/kg showed a significant and similar decrease when compared to the untreated group (*p* < 0.0001).

Since Omicron is not lethal in K18-ACE2 mice [15], the efficacy of IgG-A7 was assessed through measurements of their body weight and the viral load in their lungs. The body weights in the group infected with Omicron and treated with 0.5 or 5 mg/kg of IgG-A7 were similar to those of the uninfected control group (Figure 1D), indicating protection at those doses. In contrast, the body weights of the infected group without treatment decreased by approximately 10% on day 7 post-infection. The viral load (Figure 1E) of the group infected with Omicron and treated with 0.5 mg/kg of IgG-A7 was similar to that of the untreated controls. However, it decreased significantly in the group treated with 5 mg/kg (*p* = 0.001), indicating a protective effect at that higher dose. The efficacy of IgG-A7 at 0.5 mg/kg against Delta infection and a ten-fold higher dose (5 mg/kg) against Omicron correlated with the neutralization potency of IgG-A7 in PRNT [6], showing that IgG-A7 had a lower NC_50_ for Omicron (2.93 nM) than for Delta (0.06 nM).

### 3.2. PK Profiling

Figure 2 shows the PK curves at the single doses of 100 and 200 mg/kg. The IgG-A7 concentration in the serum as a function of time followed a typical curve of a decreasing concentration of IgG1 after the i.v. bolus administration. IgG-A7 was distributed in the body and eliminated from the blood following an exponential curve with a fast initial phase, followed by a slow elimination. The PK parameters are shown in the embedded Table in Figure 2. The 100 mg/kg dose resulted in a C_max_ of 633.36 µg/mL and an AUC of 84,294.22 µg/mL × h, whereas for the 200 mg/kg dose, the C_max_ was 1663.53 µg/mL and the AUC was 188,965.97 µg/mL × h. This difference was consistent with the higher dose of IgG-A7 for the latter. All the other PK parameters were similar for the 100 and 200 mg/kg doses, including half-lives (t_1/2_) of 266.035 and 246.691 h (11.08 and 10.28 days), k_e_ of 0.003 and 0.004 h^−1^, Cls of 0.016 and 0.017 mL/h/kg, and Vds of 5.28 mL/kg and 5.57 mL/kg, respectively. Together, these results indicated that IgG-A7 performed similarly even at 20- or 40-fold the highest efficacy dose of 5 mg/kg.

### 3.3. Tox Assessment

The potential toxic effects of IgG-A7 i.v. administration were assessed via several biochemical and hematological markers listed and described below.

#### 3.3.1. Body Weight

The body weights increased during the study as a sign of IgG-A7 tolerability and no toxicity (Figure 3A). At time zero, the male mice showed greater body weights than the females, with averages of 24.72 ± 1.6 g and 29.05 ± 1.86 g, respectively, at 100 mg/kg, and 23.13 ± 2.11 and 27.93 ± 2.02 for 200 mg/kg. The body weights at the end of the study (day 14) were 25.65 ± 2.11 g for females and 32.20 ± 2.25 g for males (100 mg/kg), whereas for females, it was 24.66 ± 2.27 and 31.33 ± 2.5 for males at 200 mg/kg. During the time of the study, neither morbidity nor mortality was observed.

#### 3.3.2. Urinalysis

The urine analysis results showed that both the control group and 100 and 200 mg/kg groups had a cloudy urine appearance with a yellow color, a density greater than ~1.050, and a pH of 6.5. Males had ~1 g/L of protein in all three groups and females had ~0.3 g/L, no glucose, no crystals, no bacterium, no ketones, and no Hb blood in all the urine determinations. All the mice had normal bilirubin and normal urobilinogen levels.

#### 3.3.3. Hematology

The results of the parameters measured for the sera from the untreated and IgG-A7-antibody-treated animals are shown in Appendix A. No changes in the samples were found compared to the reference values, control group, and groups that were administered with 100 and 200 mg/kg of the antibody.

#### 3.3.4. Blood Chemistry

The parameters of the blood tests (Appendix A) showed significant changes in the total proteins and globulins in the females and higher levels of calcium and phosphorus in both sexes. Higher levels with respect to the reference values of the males and females in both doses, however, did not vary between the control group and those administered with the antibody. No changes were shown in the other parameters measured between the controls and IgG-A7 groups at two different doses.

#### 3.3.5. Histopathology

After necropsy, the organ weights showed no statistically significant differences when compared to the control group (Figure 3B). Figure 3C shows representative micrographs of the histopathological analyses of tissue sections of the lungs, hearts, kidneys, spleens, and livers collected from the untreated mice and those treated with IgG-A7 at 100 and 200 mg/kg. The lungs showed a red coloration, without deformations in the alveoli. The hearts showed the presence of cardiomyocytes with normal nuclei located in the center of the papillary muscle and the interventricular septum. In the kidneys, no alterations in the normal structures of the proximal and distal tubules were observed in the tissues from the mice treated with IgG-A7. In the spleen tissue samples, no deformities were observed in their main structures of white pulp, red pulp, and stroma. Finally, the livers had a normal brown coloration and the presence of vacuolization (microvacuoles) with clear and round shapes.

### 3.4. Mouse and Human TCR

The preclinical safety assays of the anti-SARS-CoV-2 IgG1-A7 antibody were further investigated by evaluating the cross-reaction to mice and human tissues. To this end, 13 and 34 relevant mice and human tissues were evaluated, as recommended by the FDA and EMA [16]. Both assays employed HEK293T cells as a negative control and HEK293T cells that stably express the SARS-CoV-2 Spike protein as negative controls. As shown in Appendix A, the positive control exhibited a moderate intensity (3 within a scale from 0 to 3) and distribution of rare cell staining (++ within a scale from—to +++). No signals were detected in the negative controls. A vimentin test evinced that all the tissues employed in both assays matched the integrity criteria. In the mouse TCR assay, the IgG1-A7 antibody (0.625 µg/mL) did not cross-react with any of the evaluated tissues. Likewise, the GLP human TCR assay showed that IgG1-A7 did not bind the 36 relevant human tissues at optimal (0.625 µg/mL), supra-optimal (1.25 µg/mL), or sub-optimal (0.078 µg/mL) concentrations.

## 4. Discussion

In the previous sections, we reported that IgG-A7 protected the K18hACE2 mice from the SARS-CoV-2 Delta and Omicron variants. IgG-A7 protection against Delta correlated with a decreased viral load in the mice lungs, suggesting that its protective effect was directly related to a decrease in SARS-CoV-2 infection. For Omicron, the 0.5 mg/kg dose did not induce a significant decrease in the viral load in the mice lungs. In contrast, 5 mg/kg dose showed a significant difference with respect to the non-infected control group and the group treated with 0.5 mg/kg, suggesting that 5 mg/kg was more efficacious against Omicron than 0.5 mg/kg. Together with the previous work [6] indicating that IgG-A7 protects at 5 mg/kg against the WT strain, demonstrates that IgG-A7 is a broadly and highly effective antibody in preventing mortality in K18hACE2 transgenic mice at a dose as low as 5 mg/kg.

Having demonstrated the broad efficacy of IgG-A7 with diverse variants of SARS-CoV-2, we evaluated its PK profile in CD-1 mice. Mice, due to their low cost, rapid reproductive cycle, and ease of handling, have been used in several PK/Tox studies [17,18,19,20,21,22]. At doses of IgG-A7 20 and 40 times higher than the protective dose of 5 mg/kg, the IgG-A7 PK parameters were proportional to the dose, with a similar serum elimination half-life of ~10.5 days at both high doses, consistent with PK studies on CD-1 mice with fully human or humanized IgG1 antibodies at diverse doses [23]. For instance, a half-life of ~9.5 days has been reported for an anti-CEA IgG1 antibody and Fc fusion proteins in CD-1 mice [24]. Another report using the humanized monoclonal IgG1 antibody cantuzumab in CD-1 mice resulted in a half-life of ~6.5 days [22]. Evaluations of an anti-CD30 antibodyin other mice strains, such as Balb/c, Nude mice, and SCID, have also given half-life values in the range of IgG-A7, i.e., 7.1 ± 5 days, 10 ± 10 days, and 16 ± 8 days, respectively [25]. On the other hand, a study of COVID-HIGIV [23], a polyclonal purified human IgG product manufactured with the immunoglobulin fraction of the plasma from SARS-CoV-2 convalescent patients, at a single dose of 400 mg/kg, reported a similar half-life of 161 h (6.7 days) in healthy, wild-type C57BL/6 mice, and SARS-CoV-2-infected K18hACE2 transgenic mice. These results for t_½_, besides the low values for Vd, depict long-lasting behavior of IgG-A7 in the central compartment (blood), with a succinct distribution along other tissues.

We further evaluated the potential toxicity of IgG-A7 in the CD-1 mouse model using urine, blood, and tissue samples. At doses of 100 and 200 mg/kg, no significant differences were observed in the treated and untreated animals, indicating that IgG-A7 did not induce immune or inflammatory responses. The sera biochemical parameters, such as the total proteins, globulins, calcium, and phosphorus, showed differences in the female and male mice with respect to the reference values [26,27], which could be associated with possible renal damage [28,29]. However, when compared to the values in the control group, no statistically significant differences were found. In addition, no alterations were observed in the urea and creatinine values.

Diverse organs were also analyzed for alterations in their morphology or leukocyte infiltration. Although some tissues showed a low degree of degeneration and necrosis, this alteration could be related to the cardiac puncture and total blood loss before necropsy. However, when analyzing the kidneys, there were no changes in their weight nor structural alterations in the tissue. Therefore, changes in the total proteins, globulins, calcium, and phosphorus seemed to be related to the administration of high concentrations of a protein (IgG-A7) and its elimination via the kidneys, but did not evidence a toxic impact or histological damage in the kidney.

Finally, a mouse and human TCR study was conducted with tissues recommended by the FDA and EMA [30,31,32]. The mouse TCR study showed that IgG1-A7 did not bind the tested tissues, supporting the PK profile in the CD-1 mice, where elimination of the antibody corresponded to a typical PK profile of human IgG1 in mice, with no accumulation of the antibody in the mice organs. The TCR study on the human tissues, on the other hand, was performed under a GLP system and suggested that IgG-A7 is unlikely to have adverse effects in humans due to it having no cross-recognition with human proteins. The latter results provide support for subsequent studies on humans, serving as a bridge between the mouse studies and the eventual testing of IgG-A7 in clinical phase 1 studies. Worth mentioning is that, within the panel of human tissues, uteri (cervix and endometrium) and placenta were evaluated, which further supports the administration of IgG1-A7 to pregnant women, according to the provisions for obtaining an EUA of anti-SARS-CoV-2 antibodies.

It should be emphasized that non-human primates (NHPs) have been the primary approach for PK/Tox studies as preamble to—and in support of—clinical studies on humans. However, the urgency to develop efficacious antibodies to mitigate the devastating impact of the COVID-19 pandemic led to a high demand for and shortage of NHPs [5]. This, compounded with increasing ethical concerns [33] about the use of NHPs in preclinical studies, has required relatively inexpensive preclinical models as an alternative to NHPs, capable of providing meaningful information as quickly as possible, in order to proceed with human clinical trials. Mice are low cost, have a fast reproductive cycle, are easy-to-handle, and demand low amounts of proteins in efficacy and PK/Tox studies, thus providing expedited and useful information in numerous efficacy and PK/Tox studies [17,18,19,20,21,22]. Therefore, the information generated in this report using CD-1 mice as a PK/Tox model, complemented with the TCR study on mouse and human tissues, could be of relevance for future emerging viral infections or even non-infectious diseases, such as immunological disorders or cancer, where a fast response is needed to effectively meet unmet medical needs.

## Figures and Tables

**Figure 1 viruses-15-01733-f001:**
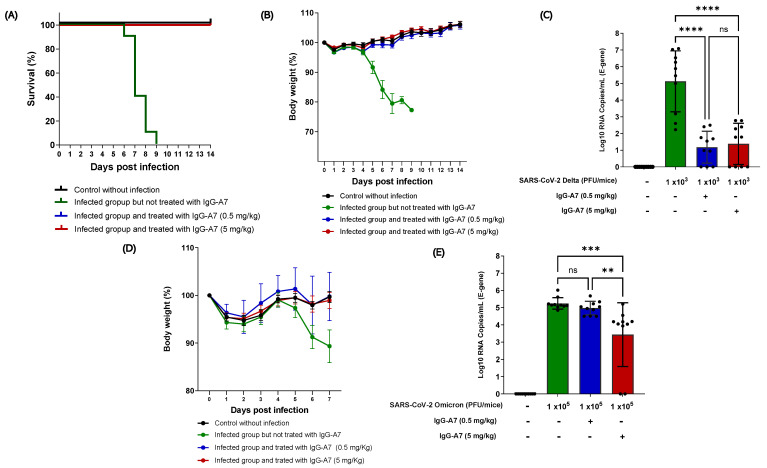
IgG-A7 protection of K18-ACE2 mice expressing hACE2 infected with SARS-CoV-2 Delta or Omicron. The efficacy of IgG-A7 against Delta variant was assessed through survival (**A**), body weight (**B**), and viral load in the lungs (**C**), whereas, for Omicron, only body weight (**D**) and viral load (**E**) were evaluated. A Kaplan–Meier statistical analysis was applied for the survival (Chi2 = 36.50, *p* < 0.0001), while a one-way ANOVA with Bonferroni’s post hoc test was applied for the viral load (**** *p* < 0.0001 for Delta and *** *p* = 0.001, ** *p* = 0.0061 for Omicron; ns = not statistically significant).

**Figure 2 viruses-15-01733-f002:**
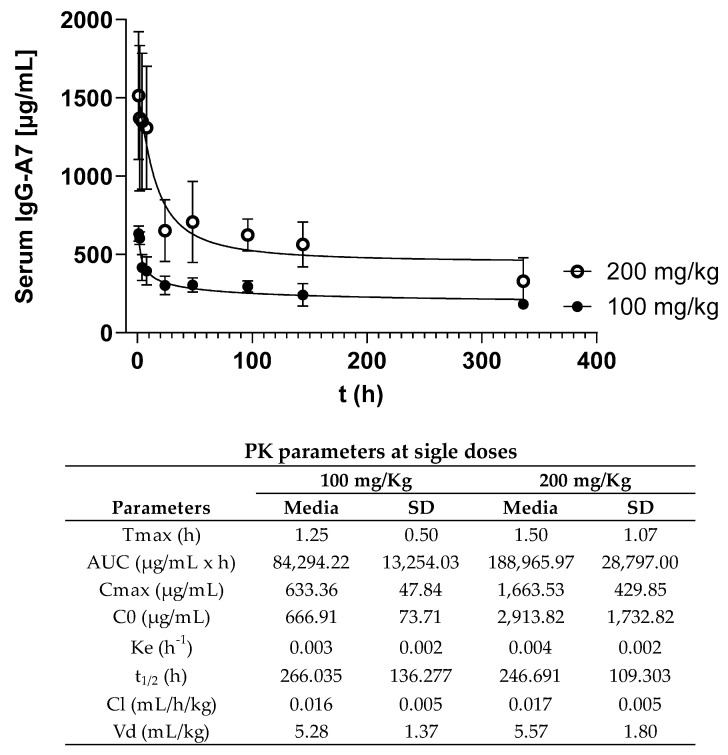
Serum concentration–time of IgG-A7 in mice serum samples at 100 mg/kg and 200 mg/kg. Each data point is the Average ± Standard Deviation (SD) of triplicate antibody quantification by ELISA of 4 mice (2 males and 2 females); n = 4. Raw data for males and females are reported in Appendix A. The table underneath the figure compiles the PK parameters calculated at 100 and 200 mg/kg single doses. Average and SD values of T_max_: time to reach C_max_; C_max_: maximal serum concentration; C_0_: estimated initial concentration; k_e_: elimination constant; t_1/2_: serum elimination half-life; AUC_0-t_: area under the curve until last sampling time; Cl: clearance; and Vd: Volume of distribution.

**Figure 3 viruses-15-01733-f003:**
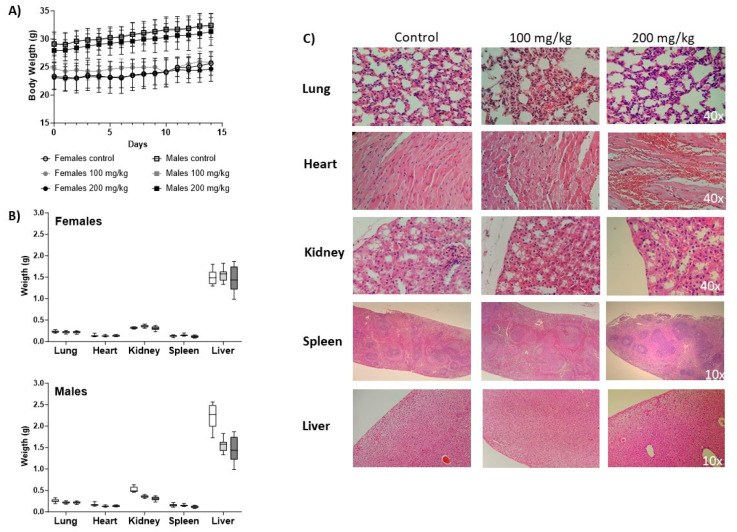
Body weight and histopathology. (**A**) Body weight measurements after IgG-A7 antibody i.v. administration at 100 and 200 mg/kg (10 males and 10 females; control group and administration group, respectively). Data are expressed as Average ± SD. (**B**) Organ weight. At the top, weight of lung, heart, kidney, spleen, and liver in females. At the bottom, weight of lung, heart, kidney, spleen, and liver in males. (**C**) Representative histological photomicrographs of stained tissue sections observed with 40× or 10×.

## Data Availability

All the data obtained during this study are included in the manuscript. Additional information could be provided by the authors upon a reasonable request.

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
