# Peer review of "Efficacy, Pharmacokinetics, and Toxicity Profiles of a Broad Anti-SARS-CoV-2 Neutralizing Antibody"

_viruses, 2023, doi:10.3390/v15081733_

Round 1
Reviewer 1 Report
This manuscript represents a significant contribution towards human health against potential future occurrence of coronavirus infections. However, at this time when the COVID-19 pandemic is fading, the chance of IgG-A7 being used clinically is rather low, so that significance of pharmacokinetics and toxicology is not obvious. In this context, the manuscript might focus more on how to streamline the protocols for the identification and optimization of efficacious antibodies in preparation for a future viral outbreak, so that a good antibody could be rapidly selected, tested, and used clinically.
In the experiments, the authors used male and female mice but there is no pharmacokinetic or toxicological result pertaining to sex differences.
In section 2.2, when five male and five female mice were given 0.5 and 5 mg/kg of IgG-A7, it is not clear who many and which sex was given 0.5 mg/kg or 5 mg/km. One could optimistically assume that five males and five females were given 0.5 mg/kg and another five males and five females were given 5 mg/km, but one could also suspect smaller sample sizes. The authors should be explicit. The subsequent descriptions are clearer. It might help to present the allocation of mice to individual experimental groups with tables.
Did some experimental mice die and consequently affect sample size? In section 2.3, when mice were sacrificed at different times after administration of IgG-A7, were mice of the two sexes randomly chosen to sacrifice or was it likely that animals that appeared sick were sacrificed first?
Did those mice that died soon after infection (6 days post infection) express more hACE2 than those that died later (9 days post infection)? It would be nice to also include a quantification of the expression of hACE2 among the experimental mice so that one can exclude the possibility that any difference in infection might be due to differential expression of hACE2 among experimental mice.
The pattern of curves in Figure 2 was somewhat unexpected. First, IgG-A7 was injected intraperitoneally, so one would expect some time before serum IgG-A7 concentration reaches the maximum. In other words, at time 0, IgG-A7 had not yet wholly gone into the blood, so the serum concentration at time 0 (and soon after) should not be at the maximum concentration. However, the serum IgG-A7 was at the maximum in both treatments. Second, the two curves look somewhat odd on right side of the chart in Figure 2. The two curves are supposed to eventually converge to the level experimentally undetectable (everything else being equal). However, the two curves are essentially parallel on the right side. It seems that the two curves were not fit with the right pharmacokinetic function. Please include the concentration-time data in a table (or a supplementary table).
English is fine.
Author Response
Reviewer:
This manuscript represents a significant contribution towards human health against potential future occurrence of coronavirus infections. However, at this time when the COVID-19 pandemic is fading, the chance of IgG-A7 being used clinically is rather low, so that significance of pharmacokinetics and toxicology is not obvious. In this context, the manuscript might focus more on how to streamline the protocols for the identification and optimization of efficacious antibodies in preparation for a future viral outbreak, so that a good antibody could be rapidly selected, tested, and used clinically.
Response:
Thank you for the comment, we fully agree with it. In fact, the closing sentence of the abstract emphasizes the point stressed by the reviewed, quoting:
“The information generated in CD-1 mouse as PK/Tox model, complemented with the mouse and human TCR, could be of relevance as alternatives to NHPs in rapidly emerging viral diseases and/or quickly evolving viruses such as SARS-CoV-2.”
Also, the closing paragraph of the discussion states:
“It should be emphasized that non-human primates (NHPs) have been the primary approach for PK/Tox studies as preamble to - and in support of - studies in humans. However, the urgency to develop efficacious antibodies to mitigate the devastating impact of COVID-19 pandemic led to a high demand and a shortage of NHPs [5]. This, compounded with increasing ethical concerns [33] on the use of NHPs in preclinical studies, has required relatively inexpensive preclinical models capable of providing meaningful information, as quickly as possible, to proceed with human clinical trials as alternative to NHPs. Mice, which are low-cost, have a fast-reproductive cycle, are easy-to-handle, and demand low amounts of proteins in efficacy and PK/Tox studies, have been providing useful information in numerous efficacy and PK/Tox studies [17-22]. The information generated in this report using the CD-1 as a PK/Tox model, complemented with the TCR study in mouse and human tissues, could thus be of relevance for future emerging viral infections or even non-infectious diseases such as immunological disorders or cancer where a fast response is needed to effectively meet unmet medical needs.”
Reviewer:
In the experiments, the authors used male and female mice but there is no pharmacokinetic or toxicological result pertaining to sex differences.
Response:
For the PK study we used 4 mice (2 male and 2 female) per time point. When compared male and females results, we didn’t see substantial qualitive differences between males and females. However, due to the small number of mice (only 2 mice per sex per time point), compounded with the inherent variability of the assay due to the difficulty to collect and process mice blood samples, we decided to pool the results of males and females to obtain a better quantitative comparison per dose. Otherwise, the results would be too noisy and, in our opinion, any quantitative conclusion derived from such noisy data could be misleading. Nevertheless, in the revised version of the manuscript we added the PK raw data for male and female serum concentrations (see Tables S2 and S3 in Supplementary Material).
In the case of Tox studies, we certainly stratified the results by male and females - See Figure 3.
Reviewer:
In section 2.2, when five male and five female mice were given 0.5 and 5 mg/kg of IgG-A7, it is not clear who many and which sex was given 0.5 mg/kg or 5 mg/km. One could optimistically assume that five males and five females were given 0.5 mg/kg and another five males and five females were given 5 mg/km, but one could also suspect smaller sample sizes. The authors should be explicit. The subsequent descriptions are clearer. It might help to present the allocation of mice to individual experimental groups with tables.
Response:
Agree. We had a Figure to visually present the experimental design but decided not to include it in the final manuscript so to minimize the number of figures. In our opinion, the following modification of lines 103-105 should clarify this point; quote: “IgG-A7 was given intraperitoneally (i.p.) in a single dose of 0.5 or 5 mg/kg 24 h post infection to groups of ten mice per dose (five male and five female).”
Reviewer:
Did some experimental mice die and consequently affect sample size? In section 2.3, when mice were sacrificed at different times after administration of IgG-A7, were mice of the two sexes randomly chosen to sacrifice or was it likely that animals that appeared sick were sacrificed first?
Response:
In the PK study (section 2.3) no mice died - see lines 115-126. We started with 72 CD-1 mice divided in two groups (36 and 36 mice, gender-balanced) and administered IgGA7 i.v. at a single bolus dose of 100 or 200 mg/kg of IgGA7. We collected blood samples by cardiac puncture at 1, 2, 4, 8, 24, 48, 96, 144 and 336 hours after administration; 4 mice (2 females and 2 males) per dose at each of the 9 time points [(9 time points x 4 mice = 36) x 2 doses = 72]. Blood samples were centrifuged at 10,000 g for 10 min at 4 °C to obtain serum. Serum samples were stored at -80 °C until analysis.
Reviewer:
Did those mice that died soon after infection (6 days post infection) express more hACE2 than those that died later (9 days post infection)? It would be nice to also include a quantification of the expression of hACE2 among the experimental mice so that one can exclude the possibility that any difference in infection might be due to differential expression of hACE2 among experimental mice.
Response:
A correlation between hACE-2 expression and survival would certainly be interesting as we can’t rule out that higher hACE2 expression led to higher infection and hence, lesser survival. However, in our opinion, such a molecular scrutiny of the mechanism of infection and subsequent protection by IgG-A7 is beyond the scope of this work.
The efficacy study was designed to assess survival due to the administration of IgG-A7. We used 10 mice gender-balanced per group to minimize biases due to experimental variability and/or variability of the animals including expression of hACE2. We performed experiments with three different SARS-CoV-2 strains. The experiments were performed in different days and settings including the previously reported results for Wuhan (Gonzales-Gonzales et al). The results were consistent across all the experiments.
In addition, we measured viral load in mice lungs to explain the mechanism of protection and found a correlation between less viral load and higher survival. In the case where we didn’t find a correlation (infection with Omicron and IgG-A7 at 0.5 mg/kg dose) we didn’t consider that as efficacious dose.
Reviewer:
The pattern of curves in Figure 2 was somewhat unexpected. First, IgG-A7 was injected intraperitoneally, so one would expect some time before serum IgG-A7 concentration reaches the maximum. In other words, at time 0, IgG-A7 had not yet wholly gone into the blood, so the serum concentration at time 0 (and soon after) should not be at the maximum concentration. However, the serum IgG-A7 was at the maximum in both treatments. Second, the two curves look somewhat odd on right side of the chart in Figure 2. The two curves are supposed to eventually converge to the level experimentally undetectable (everything else being equal). However, the two curves are essentially parallel on the right side. It seems that the two curves were not fit with the right pharmacokinetic function. Please include the concentration-time data in a table (or a supplementary table).
Response:
IgG-A7 was administered intravenously (i.v.) in the PK study - see lines 115-116, quoting “Seventy-two CD-1 mice gender-balanced were divided in two groups and administered with IgGA7 intravenously (i.v.) at a single bolus dose of 100 or 200 mg/kg of IgG-A7”
See also line 249-251, quote “Figure 2 shows the PK curves at single doses of 100 and 200 mg/kg. IgG-A7 concentration in serum as a function of the time follows a typical curve of the decreasing concentration of an IgG1 after a i.v. bolus administration.”
The serum concentration at time 0 was calculated from the parameters of the curve. The first data point was collected one hour post IgG-A7 admiration, see lines 119-122: “At 1, 2, 4, 8, 24, 48, 96, 144 and 336 hours after administration, four mice (two female and two male) at each of the nine time points were sacrificed and blood samples collected by cardiac puncture into BD Microtainer® Blood Collection Tubes (Cat. No. 365967 BD, USA).”
The fitted curves are parallel, but a close look at the raw data, shows that the last IgG-A7 concentration in serum (336 hours) at 200 mg/kg dose is close to the same data point (336 hours) at 100 mg/kg, meaning that the curves will eventually converge if more datapoints are to be collected after 14 days.
Following the reviewer’s suggestion, Tables with the PK raw data for 100 and 200 mg/kg doses are now included in Supplementary Material (Tables S2 and S3).
Reviewer 2 Report
I believe this is a very interesting and sound study reporting the results of the effect of the previously characterized recombinant anti sars cov 2 monoclonal antibody against infected transgenic mice.
The study has been planned very carefully and has been executed and scientifically explained thoroughly.
The results are very meaningful to the scientific community to the extent that the results from experiments performed on experimental animals can be applied to humans.
English language is very correctly used.
I highly recommend publication.
Author Response
Reviewer:
I believe this is a very interesting and sound study reporting the results of the effect of the previously characterized recombinant anti sars cov 2 monoclonal antibody against infected transgenic mice.
The study has been planned very carefully and has been executed and scientifically explained thoroughly.
The results are very meaningful to the scientific community to the extent that the results from experiments performed on experimental animals can be applied to humans.
English language is very correctly used.
I highly recommend publication.
Response:
Thank you for accepting the manuscript for publication and your positive and encouraging comments. Truly appreciated.